# Analysis and Research on Starch Content and Its Processing, Structure and Quality of 12 Adzuki Bean Varieties

**DOI:** 10.3390/foods11213381

**Published:** 2022-10-27

**Authors:** Lei Zhang, Weixin Dong, Yaya Yao, Congcong Chen, Xiangling Li, Baozhong Yin, Huijing Li, Yuechen Zhang

**Affiliations:** 1Hebei Province Crop Growth Control Laboratory, Hebei Agricultural University, Baoding 071001, China; 2Teaching Support Department, Hebei Open University, Shijiazhuang 050080, China; 3College of Food Science and Technology, Hebei Agricultural University, Baoding 071001, China; 4College of Agronomy and Biotechnology, Hebei Normal University of Science and Technology, Qinhuangdao 066600, China; 5College of Plant Protection, Hebei Agricultural University, Baoding 071001, China

**Keywords:** adzuki bean, DSC (differential scanning calorimetry), XRD (X-ray diffraction), in vitro digestibility, correlation analysis, cluster analysis

## Abstract

Investigating starch properties of different adzuki beans provides an important theoretical basis for its application. A comparative study was conducted to evaluate the starch content, processing, digestion, and structural quality of 12 adzuki bean varieties. The variation ranges of the 12 adzuki bean varieties with specific analyzed parameters, including the amylose/amylopectin (AM/AP) ratio, bean paste rate, water separation rate, solubility, swelling power and resistant starch (RS) content level, were 5.52–39.05%, 44.7–68.2%, 45.56–54.29%, 6.79–12.07%, 11.83–15.39%, and 2.02–14.634%, respectively. The crystallinity varied from 20.92 to 37.38%, belonging to type BC(The starch crystal type is mainly type C, supplemented by type B). In correlation analysis, red and blue represent positive and negative correlation, respectively. Correlation analysis indicated that the termination temperature of adzuki bean starch was positively correlated with AM/AP ratio. Therefore, the higher the melting temperature, the better the freeze–thaw stability. The 12 varieties were divided into Class I, Class II, and Class III by cluster analysis, based on application field. Class I was unsuitable for the diabetics’ diet; Class II was suitable for a stabilizer; and Class III was suitable for bean paste, mixtures, and thickeners. The present study could provide a theoretical basis for their application in the nutritional and nutraceutical field.

## 1. Introduction

Adzuki bean (*Vigna angularis*. L) originated in China, whose planting area and total output have always ranked top worldwide. China can produce hundreds of flavored foods and is the world’s preeminent exporter of adzuki bean, with an annual export volume of 40,000–60,000 tons. As one of the most popular beans, it has the reputation of “red pearls” in international markets; thus, it is a good source of nutrients [1,2]. This commercial crop is widely grown in East Asian countries and regions, such as China, North Korea, Japan, and South Korea. Adzuki bean seeds have high nutritional contents in various proportions, with approximately 55, 0.45, and 25% of starch, fat and protein, respectively. The kernels are also rich in iron, calcium, phosphorus, Vitamin B1, and Vitamin B2, among other minerals, as well as containing eight essential amino acids for the human body. Additionally, adzuki beans are widely employed in pharmaceuticals. Specifically, they exhibit inhibitory activity against glucosidase in streptozotocin (STZ)-induced diabetic rats [2,3].

As the main ingredient in adzuki bean, starch is an essential and inexpensive resource, composed of linear amylose and highly branched amylopectin [4]. Aside from its nutritional value, starch also serves as thickening, blending, and coating agents in food and/or non-food industries [5]. Previous studies of pulsed starch have focused on its isolation, physicochemical properties, functional properties, and modification [6,7,8]. By studying specific indicators, such as amylose content, granules and the crystal structure of different adzuki bean varieties, which may be attributed to their varying properties, the scientific basis of adzuki bean starch could be improved, thereby promoting its utilization [9]. Notably, adzuki bean starch is less digestible than cereal starch, because the high amylose content in the former leads to a high tendency toward retrogradation [10].

Resistant starch (RS) is known to resist digestion in the small intestine and further ferments in the gut. RS has been reported to reduce postprandial blood sugar-insulin and plasma cholesterol, as well as increase insulin sensitivity [11]. Fermentation of RS in the colon have been found to promote the growth of healthy gut microbes through beneficial metabolites, d reducing the incidence of certain diseases and disorders, including cardiovascular diseases, metabolic syndrome, type 2 diabetes, obesity, and colon cancer [12]. Adzuki bean starch; being a crude bean starch, has low insulin reactivity, thereby preventing chronic diseases [4,13,14]. Resistant starch (RS) is relatively difficult to degrade; it is digested and absorbed into the blood stream slowly after being ingested. As a result of a certain weight-loss effect, diabetics and other special groups can directly consume adzuki bean as health food. Colonic fermentation of RS produces many metabolites (e.g., short-chain-fat acids [SCFAs]) that are beneficial to human immunity and reduce the risk of certain metabolic syndromes, such as dyslipidemia [9]. Studies on adzuki bean starch have been particularly lacking toward further insights into the relationship between structure and functionalities [1,3,15].

During the change in thermal and crystalline properties of adzuki bean starch, the enthalpy of the starch melting endotherm is the same as the melting temperature of the crystal structure, which reflects the degree of starch crystallization; therefore, the enthalpy of fusion is usually related to the content of crystals in starch. The gelatinization temperature of starch is closely related to the X-ray diffraction (XRD) results of the starch crystal form, and it has a great relationship with the content of other processing quality indicators, such as amylopectin in starch [13,16]. Research has shown that the thermal properties of starches were influenced by various parameters, such as granule shape, amylopectin chain length, and crystalline regions [11]. In general, a crystal’s crystallinity is influenced by various factors, such as the crystal number, crystal size, and degree of interaction between a double helix and double helix orientation [17].

Most research on adzuki beans has concentrated on the functional and structural relationships of a single species. These functional characteristics include swelling power, solubility, light transmittance, freeze–thaw stability, and in vitro digestibility. Notably, their structural properties and thermal features were investigated and determined via XRD and differential scanning calorimetry (DSC) [13], respectively. There are few reports pertaining to the comparison of the processing and digestible quality of Chinese adzuki bean varieties, which are majorly cultivated in China. In addition, studies on differences in the structural properties of various adzuki bean varieties are limited [1,4,13].

The present study aimed to compare the processing quality, which entails the amylose (AM) and amylopectin (AP) content, whiteness, gel texture properties, bean paste rate, freeze–thaw stability, solubility, and swelling power, as well as the digestible quality, which involves an in vitro digestibility analysis of starch, in order to determine the thermal property and crystallization characteristics of 12 Chinese adzuki bean varieties via DSC and XRD. Furthermore, it aimed to conduct the correlation and cluster analyses of their processing, digestible, and structural activities. We originally pointed out the end-products application classifications of some specific varieties of adzuki beans. Finally, we aimed to provide a theoretical basis for the better applications of adzuki bean starch in food and non-food industries.

## 2. Materials and Methods

### 2.1. Materials

In this experiment, 12 adzuki bean varieties from different ecological regions were selected, and the varieties were selected from local varieties cultivated in different ecological regions in China. The sources, maturity and grain photo of different varieties are shown in Table 1 and Figure 1.

### 2.2. Soil Nutrient Status

The test field texture was loam, and the 0–20 cm soil composition of the test field in 2019 and 2020 were as follows: the content of alkali-hydrolyzable nitrogen; 88.37 and 84.76 mg/kg, the content of available phosphorus; 66.21 and 65.525 mg/kg, and the content of available potassium; 178.6 and 181.3 mg/kg, respectively. Fertilizer application in each plot (5 m × 1 m) was 400 g of compound fertilizer (N-P_2_O_5_-K_2_O = 24:4:8) before planting.

### 2.3. Microclimate Changes in the Community after Shading

The microclimate characteristics of the shaded area are shown as follows: the illumination intensity of 58,142.45 ± 1778.58 lx, the CO_2_ concentration (ppm) of 451.49 ± 19.63 ppm, the relative humidity of 75.51 ± 4.87%, the temperature of 26.59 ± 0.81 °C.

### 2.4. Cultivation Experiment Design

The experiment was carried out at the teaching and experimental base of Hebei Agricultural University in Baoding from 2019 to 2020 (Baoding, Hebei, 115°28′ E, 38°52′ N). Twelve adzuki bean varieties from different ecological regions in China were selected, and the natural light in the experiment season was approximately every day, with an illumination time of 14.5 h.

The experiment adopted a randomized block design, with a total of 33 plots, and each variety was repeated thrice. The plot area was 5 × 1 m, sown in holes; 2 grains per hole. The plant spacing was 25 cm, with the row spacing as 50 cm, and two rows were planted in each plot, employing 50 seedlings. The soil was loosened prior to cultivation, and after the seedlings emerged, the two true leaves were unfolded; the seedlings were set when the first compound leaves appeared. Other field management was referred to as conventional field management, which includes field weeded during the growth period, topdressing nitrogen fertilizer application (urea, content: N 46.7%, 15–20 kg/667 m^2^) and watering technique (small-scale flood irrigation) during the flowering period. Additionally, insecticide (beta-cypermethrin, content: 10%, 15–20 mL/667 m^2^) was sprayed during the flowering and podding period.

### 2.5. Starch Isolation

The adzuki bean starch was isolated by the method described by Liu et al. and Ge et al. [4,13], and was slightly modified. First, the beans were soaked in distilled water at 25 °C to soften (approximately 24 h). Thereafter, the beans were homogenized with a pulping machine (L13-Y91, Jiu yang Liability Co., Ltd., Beijing, China). The mixture was filtered through a screen of 100 μm mesh. Subsequently, the filtrate was maintained at room temperature for 6 h. Thereafter, deposited starch granules were washed with water and centrifuged at 3000 rpm for 10 min to remove impurities. Sodium hydroxide solution (0.2%) was added to eliminate the protein. After centrifugation, the protein layer on the top was scraped off, followed by thrice washing of the precipitation with water through a screen of 160 μm mesh. Subsequently, starches from adzuki bean were ground and screened through a 100-mesh sieve after drying at 45 °C (101-0A, Tianjin City Taisite Instrument Co., Ltd., Tianjin, China) for 24 h in the oven. Finally, starches from adzuki bean were stored in an airtight container before use.

### 2.6. Total Starch, Amylose (AM) and Amylopectin (AP) Content

The enzymatic hydrolysis method according to the Chinese National Standard (GB 5009.9-2016) was adopted for the determination of total starch content. The basic principle is that after removing fat and soluble sugars from the sample, starch is hydrolyzed into small molecular sugars with amylase (Gao Feng’s amylase: enzyme activity ≥ 1.6 U/mg, Beijing Solarbio Technology Co., Ltd., Beijing, China), followed by hydrolyzing into glucose with hydrochloric acid (6 mol/L), and finally determined with the content of glucose. Then, 5.00 mL of alkaline copper tartrate solution A (15 g of copper sulfate and 0.050 g of methylene blue were dissolved in water and fixed to 1000 mL) and 5.00 mL of alkaline copper tartrate solution B (50 g of sodium potassium tartrate and 75 g of sodium hydroxide were dissolved in water, and then, 4 g potassium ferrocyanide was added and fixed to 1000 mL with water) were placed in a 150 mL conical flask, and then 10 mL of water was added. Two glass beads were put into the mixture, and about 9 mL of glucose standard solution from the burette was dropped, controlling the mixture to boil within 2 min. The solution was kept boiling, continuing to drop glucose at the rate of one drop every two seconds until the blue color of the solution faded as the endpoint, recording the total volume of glucose standard solution consumed, and making triplicate, taking the average value to calculate every 10 mL (5 mL for solution A and 5 mL for solution B) of alkaline copper tartrate solution equivalent to the mass of glucose, m_1_ (mg). The glucose content (X1) in the sample was calculated according to the following formula, which was then converted to starch content [18].
(1)$$X_{1}=\frac{m_{1}}{\frac{50}{250}\times \frac{V_{1}}{100}}\;\;\;\;\;\;\;\;X=\frac{{(X}_{1}-X_{0})\times 0.9}{m\times 1000}\times 100$$

In the formula: X is the content of starch in the sample (g/100g); X_1_ is the amount of glucose in the sample (mg); X_0_ is the reagent blank (mg); m is the sample weight (g); m_1_ is a 10 mL alkaline copper tartrate solution that is equivalent to the mass of glucose (mg); V_1_ is an average volume of sample solution consumed during determination (mL).

The Chinese national standard method (GB 7648-87) was adopted for the determination of amylose. The basic principle entails that starch and iodine form an iodine–starch complex, which has a special color reaction. Amylopectin and iodine form a brown-red complex, while amylose and iodine form a dark blue complex. Under the condition that the total amount of starch remains unchanged, the two starch dispersions are mixed in different proportions, and react with iodine under certain wavelength and acidity conditions to generate a series of colors, from purple red to dark blue, representing their different amylose and branched starch. Amylose content ratio, according to absorbance and amylose concentration, has a linear relationship, which can be measured by spectrophotometer (Model: 752N, Manufacturer: Shanghai INESA Analytical Instrument Co., Ltd., Shanghai, China).
(2)$$\mathrm{Amylose}(\%,\mathrm{percentage}\;\mathrm{of}\;\mathrm{total}\;\mathrm{starch})=\frac{G\times 100}{m_{1}\times 5}\times 100$$

In the formula: G is amylose content calculated by regression equation, mg. m_1_ is the mass of crude starch contained in the weighed sample, 100 mg.

### 2.7. Starch Granules Size

The size of starch granules was measured with a micrometer under an optical microscope (M = 100×; Olympus BX51 optical microscope, Shanghai Yuehe Biotechnology Co., Ltd., Shanghai, China). Four effective fields of view were selected (at least five starch granules in each field) from the starch granule samples of each variety, followed by measuring the length and width of the largest starch granule in each effective field of view, using the starch granule length (×1, μm) and the geometric mean of the width (×2, μm).

### 2.8. Whiteness

The determination method employed for whiteness evaluation was consistent with the Chinese national standard (GB/T 22427.6–2008). The whiteness of adzuki bean whole powder was measured using an intelligent whiteness tester (WSB-VI, Daji Photoelectric Instrument Co., Ltd., Hangzhou, China). The intelligent whiteness tester was calibrated using a white tile (Y = 92.3, x = 0.3156, y = 0.3433). The whiteness values were measured thrice, and an average value was calculated. Several studies regarding the evaluation of whiteness have been reported for various food substances, such as cassava raw materials, avocado seed starch, rice, wheat, etc. [19,20].

### 2.9. Texture Properties

Adzuki beans were grounded with a 300 W grinder and then passed through a 40-mesh sieve to obtain adzuki bean flour. Subsequently, 50 g of adzuki bean flour was placed in a glass dish (diameter 10 cm, height 1.5 cm), followed by mixing with 20 mL of distilled water. The adzuki bean flour mixture was steamed for 45 min in an atmospheric pressure hot water boiler (DJDG-24 3L, Tai’an, China), and cooled at room temperature. The texture characteristics analysis of adzuki bean gel was conducted, adopting the TMS-PRO Food Property Analyzer (FTC Company, Sterling, VA, USA). The texture analysis parameters were set as follows: the test speed was 1 mm/s, deformation was 50%, and the starting force was 0.5 N. Notably, each sample was measured 6 times. The maximum and minimum values were removed, and the average values were taken as the final test result. From the texture profile curve, the following parameters were calculated: hardness, cohesiveness, adhesiveness, springiness, and chewiness.

### 2.10. Bean Paste Rate

The adzuki bean paste was isolated by the method adopted by our predecessors in this research field [21], with a slight modification. The total mass of the adzuki beans placed in the aluminum box was weighed (M, g). Thereafter, it was soaked in distilled water at room temperature for 10 h. These beans were collected and steamed for 1 h in an atmospheric pressure hot water boiler. The cooked beans were poured into a mortar and were thoroughly pestled. After being ground, the paste was washed by a 60-mesh sieve. The sediments were collected and dried in a blast dryer at 80 °C for 48 h to obtain dry bean paste (W_0_, g). The bean paste rate (C, %) was calculated according to the following formula:(3)$$C(\%)=\frac{W_{0}}{M}\times 100$$

### 2.11. Freeze–Thaw Stability

The empty 10 mL centrifuge tube with a lid was weighed (m_0_). Subsequently, 300 mg of adzuki bean starch was dissolved in 5 mL of distilled water to prepare the starch suspension. The suspension was kept at 100 °C for 30 min. Thereafter, it was cooled at room temperature and put in a freezer at −20 °C for 24 h. The suspension was taken out, thawed naturally at room temperature, and weighed (m_1_). The suspension was centrifuged at 3000× *g* for 15 min. The supernatant was discarded, and the sediment with centrifuge tube was weighed (m_2_). The water separation rate was calculated as follows:(4)$$\mathrm{Water}\;\mathrm{separation}\;\mathrm{rate}\;I=\frac{{(m}_{1}-m_{2})}{{(m}_{1}-m_{0})}\times 100$$

### 2.12. Solubility and Swelling Power

Solubility and swelling power tests were conducted according to Molavi, Razavi, and Farhoosh [22], and slightly modified. Notably, the empty 10 mL centrifuge tube with a lid was weighed prior to the experiment.

Thereafter, 100 mg of adzuki bean starch was dissolved in 5 mL of distilled water to prepare the starch suspension. The suspension was kept at 90 °C for 30 min, with subsequent stirring at every 2 min. Following stirring, it was cooled at 25 °C and centrifuged (3500× *g*, 15 min). The supernatant was collected in the pre-numbered constant weight aluminum box and placed in a drying oven at 105 °C to dry to a constant weight. Finally, the box was taken out and cooled at room temperature for being weighed.
(5)$$\mathrm{Solubility}\;(\%)=\frac{\mathrm{Water}-\mathrm{soluble}\;\mathrm{starch}\;\mathrm{weight}}{\mathrm{Starch}\;\mathrm{sample}\;\mathrm{weight}\;(\mathrm{dry})}\times 100$$
(6)$$\mathrm{Swelling}\;\mathrm{power}\;(\%)=\frac{\mathrm{weight}\;\mathrm{of}\;\mathrm{sediment}}{\mathrm{Starch}\;\mathrm{sample}\;(\mathrm{dry})\times (100-\mathrm{Solubility})}\times 100$$

### 2.13. In Vitro Digestibility Analysis

This analysis was performed according to the method adopted by Hu et al., Xu et al. and Ge et al. [1,6,13], among other enzymatic hydrolysis methods, with slight modifications. Each starch sample (200 mg) was suspended in a bottle with a sodium acetate buffer (15 mL, 0.1 M, pH 5.2). Thereafter, the bottle was heated in boiling water for 15 min. The starch dispersion was cooled at 37 °C and mixed with multiple enzymatic solutions (10 mL, containing 140 U amyloglucase and 290 U porcine pancreatic alpha amylase). Subsequently, the mixture was incubated and shaken in a 37 °C bath (150 rpm). Complete subdivision (0.5 mL) was administered at different intervals (10, 20, 30, 40, 60, 90, 120, and 180 min) and mixed with 4 mL absolute ethanol (99.7%, *v*/*v*) to inactivate the enzyme. The digestibility of starch was determined by 3,5-dinitrosalicylic acid (DNS). Based on the rate of hydrolysis, starch was defined as rapidly digestible starch (RDS, digested in 20 min), slowly digestible starch (SDS, digested within 20–120 min), and resistant starch (RS, digested in 120 min). The RDS, SDS, and RS fractions were determined by the following formulas:RDS(%) = 0.9 × Gt(7)
SDS(%) = 0.9 × (G120 − G20)(8)
RS(%) = T − RDS − SDS(9)

In the formula: Gt denotes the reducing sugar content at each interval. Accordingly, RDS, SDS, and RS are the rapidly digestible starch, slowly digestible starch, and resistant starch, respectively. Lastly, T expresses total starch weight.

### 2.14. Thermal Properties

Thermal properties were conducted according to Ge et al. [13] and Hong et al. [23] and were slightly modified. Thermal properties of adzuki bean starch were analyzed using differential scanning calorimetry analyzerQ2000 (Waters Technology (Shanghai) Co., Ltd., Shanghai, China) with a Refrigerated Cooling Accessory from American TA Instruments compony. First, 5 mg of sample was put in an aluminum dry pan, a certain amount of deionized water was added and mixed well (sample: water = 1:2), sealed and covered, then taken out after in a refrigerator at 4 °C for 24 hours later, and a certain amount taken. The sample was placed into the crucible, sealed and placed in the instrument, the program set, and the differential scanning calorimetry analyzer to was used to measure. The empty dry pot was the control, the heating rate was 10 °C/min, the temperature range was 20~110 °C, nitrogen flow rate 50 mL/min, the DSC thermal effect curve was obtained, and the values of onset (To), peak (Tp), conclusion (Tc) and enthalpy of gelatinization (ΔH) were calculated for characteristic parameters by using the software of Origin 2021.

### 2.15. Changes in Crystallization Characteristics (XRD)

The X-ray diffraction pattern (XRD) was measured using the Bruker D8 Advance instrument produced in Germany. A fully water-balanced starch sample (equilibrated for 72 h in an environment with a relative humidity of 100%) was tightly packed into a plastic abrasive tool (diameter 20 mm, depth 1.5 mm), and measured under X-ray. The determination was operated at 40 kV and 40 mA with Cu Kα radiation (λ) at 0.1542 nm. XRD patterns were acquired for a 2θ range of 4~45° with a step size of 0.02 and a step rate of 3/min. The degree of crystallinity was calculated according to the method of Zeng et al. [7,8,11].

### 2.16. Correlation and Cluster Analysis of Each Indicator

All data were processed by Origin (version 2021, OriginLab compony, Northampton, Massachusetts, the United States). All values were expressed as mean ± SD. Notably, SPSS (version statistics 24, IBM company, Amunk, NY, USA) software was used for correlation and cluster analysis. Among them, cluster analysis adopts the method of hierarchical cluster, adopts between-groups linkage, and the measurement interval is squared Euclidean distance. Correlation analysis was performed using bivariate correlation, and the correlation coefficient was Pearson. Furthermore, Dunnett’s multiple range tests were used to determine the significant differences between a group means at *p* < 0.05. 

The starch granules were observed, measured, and photographed by Olympus BX51 optical microscope. Lastly, MDI Jade (version 6, The International Centre for Diffraction Data, Newtown Square, PA, USA) software was adopted for the calculation of crystallinity.

## 3. Results and Analysis

### 3.1. Amylose (AM) and Amylopectin (AP) Content, AM/AP Ratio, Starch Size, Total Starch Content and Total Protein Content

The contents of amylose (AM) and amylopectin (AP) are respectively shown in Figure 2A,B. Among the adzuki beans varieties, the amylose content of Tangshanhong ranked amylose at the top at 28.02%, and Baihong 2 reached the valley at 5.22%. The average value of the aforementioned varieties was 13.81%, while the average amylopectin content was 86.19%. The highest and lowest amylopectin content were 94.77% (Baihong 2) and 71.98% (Tangshanhong), respectively. Among the evaluated AM/AP ratios *(*Figure 2C), those of the varieties ranged from 5.52% (Baihong 2) to 39.05% (Tangshanhong). Notably, the average AM/AP ratio was calculated as 16.53%.

The content of amylose in adzuki beans varies depending on the variety. Previous studies have reported that the native red adzuki bean starch had apparent amylose contents of 24.33% to 26.64% [4]. Our AM and AP data showed some disparity from the previous results, and these may result from the differences in the tested varieties and crop cultivation conditions [13]. Notably, the AM/AP ratio had a significant effect on the texture properties, among other processing qualities of starch of different adzuki bean varieties; thus, this indicator drew much attention from the breeders, researchers, and processors.

As shown in Figure 2D, consequent to the evaluation of starch granule sizes of various adzuki bean varieties, the largest and smallest values were respectively obtained as 49.79 of Pinhong 2011 and 35.03 of Tongza 6.

The biosynthesis of starch is a complex biochemical process, and a series of enzymes are involved in the formation of starch granules. The biosynthesis of AP was mainly composed of soluble starch synthases (SS), which were catalyzed by starch-branching enzymes (SBEs) and starch-debranching enzymes (DBE), while AM was mainly catalyzed by granule-bound starch synthase (GBSS) encoded by waxy genes catalytic synthesis [24].

The photosynthetic products of leaves mainly exist in the form of sucrose and were translocated into the grains, followed by degradation to generate uridine diphosphate glucose (UDPG) and fructose before they could be used to synthesize starch. Illustratively, the synthesis of sucrose was controlled by sucrose phosphate synthase (SPS), while an enzyme–sucrose synthase (SS) catalyzed the sucrose degradation. Finally, under the action of these starch synthases, the synthetic starch was stored in the grain [25].

The differences in starch granule content and starch granule size distribution among the adzuki bean varieties were mainly due to different genotypes, as well as differences in various cultivation parameters, including field temperature, light, water, fertilizer, among others. Adzuki bean is a typical short-day crop, such as Baihong 2, which is planted in Baoding. Invariably, these species are poised with shortened sunshine hours during the day, resulting in insufficient vegetative growth and accelerated reproductive growth. Shading can improve the activities of SS and SPS enzymes and can strengthen the synthesis of starch. Per the total starch content data, the northern variety Baihong 2 (31.258%) had significantly higher values than that of the southern variety Yuhong 2 (29.776%). Notably, more research results reported in this experiment further clarified the differences in AM and AP contents in total starch of various adzuki bean varieties. Although the research on starch metabolism pathways has been widely conducted, with the involved enzymes and genes explicitly elucidated [24], the mechanism of the interaction of these enzymes in the starch synthesis pathway, signal transduction, and starch metabolism regulation still require further investigation.

### 3.2. Whiteness of Adzuki Bean Crude Starch and Texture Properties of Gels Processed by Adzuki Bean Flour

From Table 2, the least whiteness of crude starch flour among the adzuki bean varieties was evaluated as Jihong 1 (118.03), and the highest was determined to be Pinhong 2011 (141.33), with an average value of 131.80. Customarily, CIE L value is used to characterize the whiteness of starch. However, in this experiment, the intelligent whiteness tester (2.8) was adopted to measure the starch whiteness among different adzuki bean varieties, with reference to the Chinese national standard. The instrument is quite simple and convenient for application, since it is calibrated and ensures direct reading, instead of calculating the result from the brightness, redness, and yellowness values. Compared with other crops, such as wheat and rice, the whiteness and appearance quality of the crude starch of adzuki bean is better [19,21,26,27].

Commercially, adzuki beans are mainly employed for their desirable gel quality; therefore, the gel produced by adzuki bean flour was determined. Among the various indicators of texture properties, the hardness of each adzuki bean variety varies from 56.23 to 176.33 N; with the tenderest evaluated as Baihong 2, while the hardest is determined as Jihong 9218. The variation range of cohesion was between 0.06 and 0.19; with the least determined as Baihong 2, while the greatest was Tangshanhong. The variation range of elasticity sets between 0.56 and 2.79 mm; with the lowest being Baihong 2, while the greatest was Tangshanhong. The variation range of adhesiveness was between 3.38 and 34.07 N, with the least determined as Baihong 2 and the greatest being Jihong 9218. The variation range of chewiness sets between 2.35 and 88.98 mJ; with the least evaluated as Baihong 2, and the highest being Tangshanhong. The differences between the indicators of varieties are shown in Table 2.

The study found that flour processed from adzuki beans was less cohesive, firmer, and chewier than lupin (*Lupinus albus* L.) flour, as measured by texture indices [28].

### 3.3. Bean Paste Rate, Freeze-Haw Stability, Solubility, Swelling Power and In Vitro Digestibility Analysis

Bean paste rate is one of the most important processing quality traits of adzuki beans. A bean paste substance entails several gelatinized and swollen starch granules, with a size of approximately 60–300 mesh [15]. From Figure 3A, the bean paste rate of each adzuki bean variety varied from 44.7 to 68.2%. Moreover, Baohong 876 has the highest paste production rate, while Henong 3 indicates the weakest rate. While cooking the bean paste, starch and water were first mixed together, followed by the subsequent absorption of water by the starch particles, which invariably expand; when they were heated, the starch molecules began to violently expand. When the starch was wobbled, the intramolecular and intermolecular hydrogen bonds were broken, resulting in the destruction of the original crystal structure of starch. Thereafter, starch granules were wrapped by the thermal coagulation of the proteins existing in the cell body before starch gelatinization. Regarding the total starch content *(*Figure 2E) of Baohong 876 (30.931%) and Henong 3 (30.308%), as well as the protein content *(*Figure 2F) of Baohong 876 (28.102%) and Henong 3 (25.983%), the contents of protein and starch were directly related to the viscosity and yield rate of adzuki bean paste. This has similar results in mung beans [21].

The freeze-thaw stability of starch paste was the key aspect affecting the quality of starch-based frozen foods, which was a measurement of starch ability to withstand adverse physical changes caused by freezing and thawing processes [4]. Figure 3B shows that the water separation rate of freeze-thaw stability varies from 45.56% (Henong 1) to 54.29% (Tongza 6). Regarding freeze-thaw stability, similar results have also been found in other starch crop studies. Specifically, the water separation rate of starch in rice sets between 30 and 35% [29]. Depending on the number of cycles, that of corn starch sets between 25–55% [30]. The freeze-thaw stability also reflects the water-holding capacity of starch molecules, and the greater the water separation rate of starch, the worse the freeze-thaw stability. Therefore, it has a close relationship with the molecular structure of starch. The amylose ratio, interaction between starch chains, and spatial structure of the molecule played vital roles toward affecting the freeze-thaw stability of starch [31].

The solubility (Figure 3C) of different adzuki bean starches varied from 6.79% (Liaohong 08712) to 12.07% (Baihong 2). The variation range of swelling power (Figure 3D) was 11.83% (Baihong 2) ~15.39% (Jihong 9218). The results of this study are basically consistent with the results of Liu et al. [4].

Solubility can reflect the interaction force between starch granules and water molecules. The dissolution of starch mainly resulted from the escaping of amylose and small amylopectin off the swollen granules. As the temperature increased, water molecules permeated into the starch granules, invariably resulting in swelling of the starch granules. Meanwhile, the uncrystallized portion of the amylose gradually dissolved in water due to heating, thereby increasing the solubility of starch. Dense accumulation of starch in amorphous regions, particle integrity and amylose interactions may be important factors affecting starch expansion and solubility [13].

The swelling characteristic was the dynamic process of starch gelatinization. The swelling reaction of starch reflects the main characteristic of amylopectin, strength of starch crystallization, and compactness of starch granules [4,13,32].

As seen in Figure 4, the various starch contents of different adzuki bean varieties showed that the content of RDS in adzuki bean starch was the highest, followed by SDS, while RS content was the least. The range of RDS sets between 46.352% (Jihong 9218) and 63.422% (Henong 3). Several studies have shown that RDS was the most important factor for postprandial blood sugar increase [6,10]. The slowly digestible starch (SDS) and resistant starch (RS) have relatively low glycemic response, which contributes in the mitigation of those metabolic diseases. In the results of this experiment, SDS content ranged from 13.192% (Liaohong 08712) to 41.628% (Jihong 9218) among all evaluated varieties, whereas the RS content sets between 2.020% (Jihong 9218) and 14.634% (Liaohong 08712). This is basically consistent with the data of previous studies [13].

Adzuki bean is a cereal crop with the same origin as medicine and food, and it has certain medicinal and health care functions. The amount of starch, RS and SDS contents could be employed as an indicator for healthy food to an extent, among special groups.

### 3.4. Starch Crystallization Characteristics (XRD)

The crystalline structure is an important factor affecting the function of starch [4]. X-ray diffraction was widely used to study the crystalline structure of starch [9,13,14]. According to the X-ray diffraction pattern, the crystalline structure types of starch granules can be divided into type A, B, and C. The structure and function of starch granules of different crystalline types varied apparently. Among them, the type C crystal was an intermediate state continuously changing from type A to type B and was composed of type A and type B crystals. Notably, type C can be transformed from type A or type B under specific or predetermined conditions. Therefore, type C can also be regarded as a mixture of type A and type B, which was mainly distributed in beans and rhizomes of yam plants. The contents of type A and type B crystals in type C crystals varied. The spectra were also different and could be further divided into type C, type CA (mainly A-type crystal), and type CB (mainly B-type crystal) [4]. Furthermore, type C consisting of minute starch granules can significantly increase the intensity of scattering peaks, solubility, gelatinization temperature, viscosity coefficient, degree of hydrolysis, digestibility, among others [14]. Starches of different crystalline types have distinct characteristic peaks as follows: type A has three strong peaks at 15°, 17°, 18°, and 23°; type B has strong diffraction peaks at 5.6°, 17°, 23°, and 24°; type C with a combination of type A and type B, and has a moderately strong peak at 5.6° when compared with type A, which might disappear in dry or partially dried samples, whereas in the comparison with type B, it showed a single peak at 23° [13]. Among the 12 evaluated varieties (Figure 1), Tongza 6 had the highest crystallinity (Figure 5) of 37.38%, while Baohong 876 exhibited the least with 20.92%. The X-ray diffraction pattern of adzuki bean starch (Figure 5) showed that it belonged to type CB (mainly type B crystals) at 15°, 17–18°, and 23°. Strong diffraction peaks can be observed at 15°, 17–18°, and 23°. Accordingly, Xu et al. [1] reported that the native red adzuki starch presented a typical type C pattern, which showed intense crystallographic peaks at specific diffraction angles of 15°, 17°, and 23°. Similar to the above study [4,13,14], the native red adzuki bean starch exhibited a type C crystalline structure, i.e., a mixture of type A and B crystalline structures.

### 3.5. Thermal Properties

During water heat treatment, starch gradually expands, which invariably cleaves the hydrogen bonds within and between molecules. Consequently, starch molecules diffuse, followed by induced conformational changes of the polysaccharide units, transforming from an ordered crystalline phase to an amorphous phase. The energy changes in this process can be described in DSC curves [13].

In the thermogram obtained by DSC, the starting temperature T_o_ represents the melting temperature of the most unstable starch crystals, the peak temperature T_p_ denotes the melting temperature of most starch crystals, and the final temperature T_c_ implies the melting temperature of the most complete starch crystals. Enthalpy is an important characteristic parameter of DSC, representing the energy consumed by starch molecules to melt hydrogen bonds and maintain starch in a water-soluble state [9,13]. However, enthalpy can also indicate aging. The energy absorbed by the starch recrystallization during melting and its magnitude reflects the overall crystallinity of the starch [13,14].

The various indicators of the starch thermodynamic determination of each adzuki bean variety are shown in Table 3. Illustratively, T_o_ was from 74.01 (Jihong 9218) to 54.06 °C (Baohong 876), with an average of 61.91 °C and had standard deviation of 5.557 °C. Furthermore, Baohong 876 presented the highest T_p_ (78.76 °C) among all the evaluated varieties, while Henong 3 had the lowest T_p_ (69.52 °C). The average value of T_p_ was 74.21 °C, and the standard deviation was calculated as 3.353 °C. The variation range of Tc sets was between 81.83 (Jihong 1) and 92.71 °C (Tangshanhong). The average value of T_c_ was 85.15 °C, and the standard deviation was evaluated as 3.809 °C. The highest enthalpy (5.80 J/g) was found in the Tongza 6 variety, while the lowest was presented by Tangshanhong (3.43 J/g), with an average of 3.43 J/g, and standard deviation determined as 0.987 J/g. The data of T_o_, T_p_, and T_c_ in this experiment are basically consistent with the research results of Honda et al. [14] and Xu et al. [1]; however, the enthalpy ΔH results varied.

### 3.6. Correlation and Cluster Analysis of Each Indicator

In this study, 12 adzuki bean varieties from different ecotype regions were selected, and their starch solubility, swelling power, freeze-thaw stability, amylose and amylopectin content, AM/AP ratio, and texture properties were individually evaluated. The correlation analysis (Figure 6) revealed that the bean paste rate was positively correlated with the contents of total starch, soluble protein, and amylopectin. This implies that the bean paste content was related to the content of starch and protein in adzuki beans. The five indicators of texture properties were positively correlated with AM/AP ratio, among which elasticity, adhesiveness, and chewiness were significantly positively correlated. For amylose content, adhesiveness and chewiness were significantly positively correlated with AM/AP ratio. The reason for this gel texture property result may be that after starch gelatinization and cooling to form a gel, starch molecular chains were cross-linked and polymerized by hydrogen bonds [33]. The higher the straight-chain content, the more hydrogen bonds that were generated. Finally, it reflected in good textural properties, such as hardness, elasticity, adhesiveness, and chewiness [7]. The structural damage among the various starches varied, and the precipitation of water was different, which may lead to the differences in freeze-thaw stability of various starches.

The solubility was negatively correlated with the amylose content and AM/AP ratio. This may be attributed to the dissolution of starch granule crystals and the invariable dissociation of amylose during starch swelling. Swelling power was positively correlated with the amylose content and AM/AP ratio, had an insignificant negative correlation with amylopectin content and crystallinity, and was significantly negatively correlated with freeze-thaw stability and RDS. This may be attributed to the fact that as the temperature approaches the gelatinization temperature of starch, the microcrystalline bundle structure of starch begins to loosen, for the exposed polar groups of starch to combine with water, resulting in the rapid absorption of surrounding water by the starch granules, and invariably increasing its expansion potential [23].

Correlation analysis (Figure 6) showed that the termination temperature T_c_ was significantly positively correlated with AM/AP ratio, indicating that T_c_ was closely related to the structure and composition of starch. Both T_c_ and T_p_ indicated a very significant positive correlation, and their trend among the varieties showed similar changing alteration patterns, while T_o_ had a significant positive correlation with the freeze-thaw stability, indicating that the better the freeze-thaw stability, the higher the melting temperature of the starch crystals.

Cluster analysis was performed on a total of 22 indicators to obtain the final pedigree map. As shown in Figure 6, where the squared Euclidean distance was 10, the 12 adzuki bean varieties were divided into three categories as follows: Class I included Henong 1, Henong 2, Henong 3, Liaohong 08712, Tongza 6, Pinhong 2011, Jihong 1; Class II contained Tangshanhong and Jihong 9218; and lastly, Class III consisted of Baohong 876, Yuhong 2, and Baihong 2.

As shown in Appendix A, Class I adzuki bean varieties had relatively higher solubility and enthalpy values than those of Class II and Class III. Therefore, Class I adzuki bean varieties had relatively high RDS content, which is unsuitable for the dietary needs of diabetics. Class II adzuki bean varieties had the highest swelling power, freeze-thaw stability, and AM/AP ratio. In addition, the five indicators of texture properties, SDS contents, T_o_ and T_p_ were all relatively high. Meanwhile, the degree of crystallinity sets between Class I and Class III. Therefore, Class II adzuki bean varieties were suitable as stabilizers, belonging to gel-type food additives, which can improve the texture properties of foods. This is conventionally applied in dairy products and sausages. Compared with the other two types, Class III adzuki bean varieties had relatively high amylopectin content, paste yield, T_c_, and crystallinity. Hence, Class III varieties were suitable for making bean paste food. Moreover, the high content of amylopectin was suitable for application in food processing, serving as a water retention agent or as a sustained release agent, binder, and thickener in other aspects, and it was relatively easy to be gelatinized.

## 4. Conclusions

The DSC conclusion temperature of adzuki bean starch was significantly positively correlated with AM/AP ratio, indicating that the higher the melting temperature of the crystals, the better the freeze–thaw stability of adzuki bean starch.

The crystallinity varies from 20.92 to 37.38%. According to the analysis of X-ray diffraction pattern, strong diffraction peaks appear at 15°, 17°, 23°, which confirmed that the crystal types of adzuki bean starch belonged to type BC (mainly B-type crystals).

Cluster analysis was performed on a total of 24 indicators, and 12 different adzuki bean varieties were divided into Class I, II, and III as follows: Class I included Henong 1, Henong 2, Henong 3, Liaohong 08712, Tongza 6, Pinhong 2011, Jihong 1, with lower SDS content and higher RDS content, which are unsuitable for the dietary requirements of diabetics; Class II contained Tangshanhong and Jihong 9218, which were suitable to be added to gel food as a stabilizer, and thus improve the texture characteristics of food, and they could be employed in the processing of dairy products and sausages; lastly, Class III consisted of Baohong 876, Yuhong 2, and Baihong 2, which were suitable for bean paste food with high throughput and good commerciality, as well as for mixtures and thickeners owing to its relative potential for easy gelatinization.

Overall, this study plays a positive guiding role toward the improvement of adzuki beans breeding quality, which can provide important reference significance for the development of adzuki bean variety resources and further present a theoretical basis for the application of adzuki bean starches in food and non-food fields.

## Figures and Tables

**Figure 1 foods-11-03381-f001:**
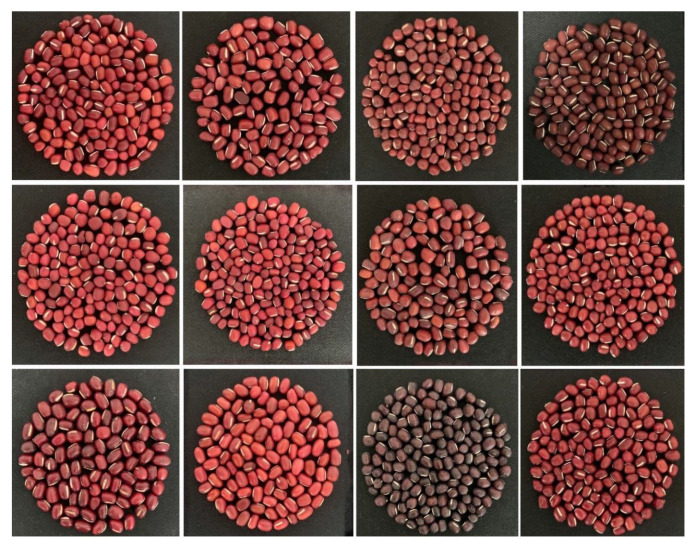
Grain photos of the tested adzuki bean varieties.

**Figure 2 foods-11-03381-f002:**
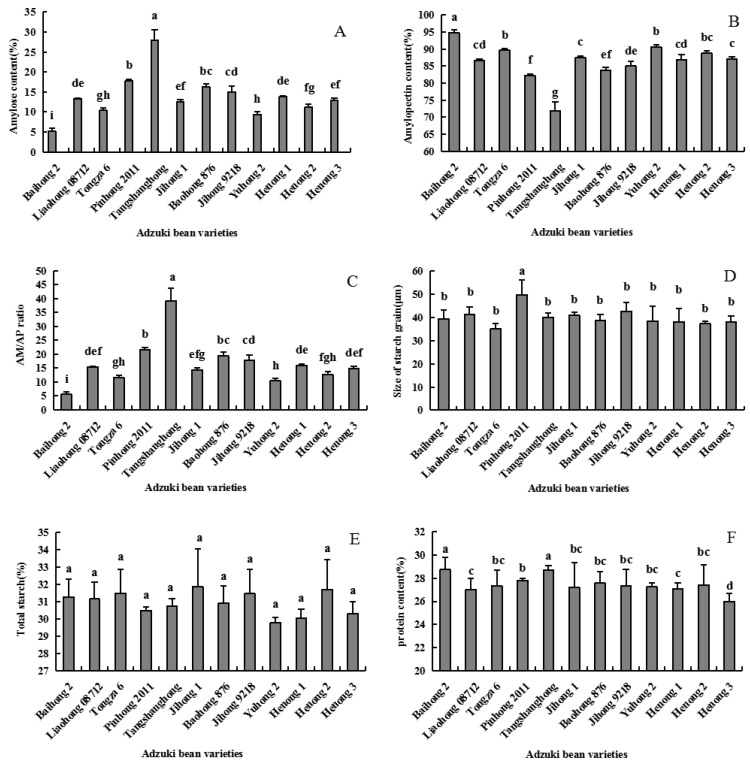
The AM (**A**) and AP (**B**) content, AM/AP ratio (**C**), starch size (**D**), total starch (**E**) and protein content (**F**) of different adzuki beans. Note: Different lowercase letters indicate significant difference of 0.05 level, LSD method.

**Figure 3 foods-11-03381-f003:**
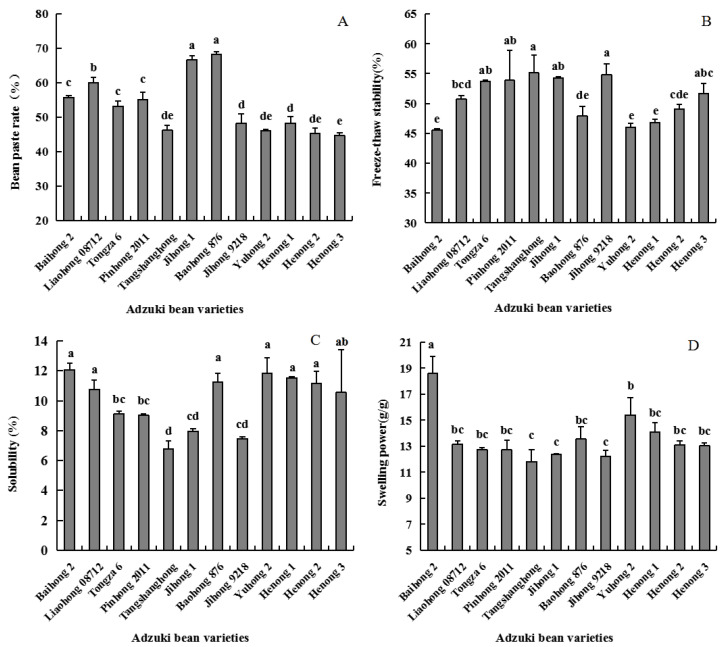
The bean paste rate, freeze-thaw stability, solubility, and swelling power of different adzuki bean varieties. Note: Different lowercase letters indicate significant difference of 0.05 level, LSD method. (**A**): Bean paste rate of adzuki bean flour (%); (**B**): Freeze-thaw stability of starch (%); (**C**): Solubility of starch (%); (**D**): Swelling power of starch (g/g).

**Figure 4 foods-11-03381-f004:**
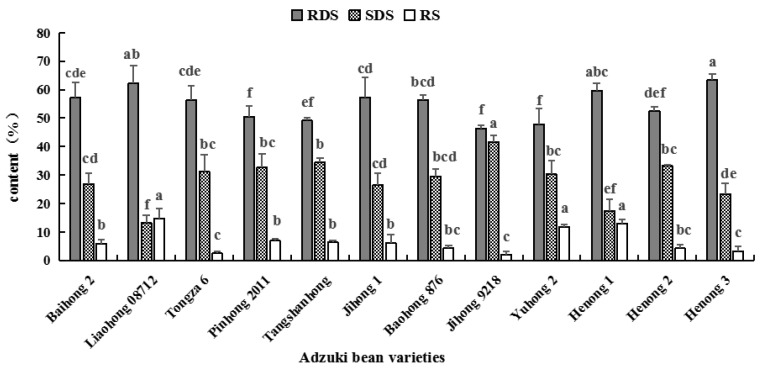
Contents of RDS, SDS and RS in starch. Note: Different lowercase letters indicate significant difference of 0.05 level, LSD method.

**Figure 5 foods-11-03381-f005:**
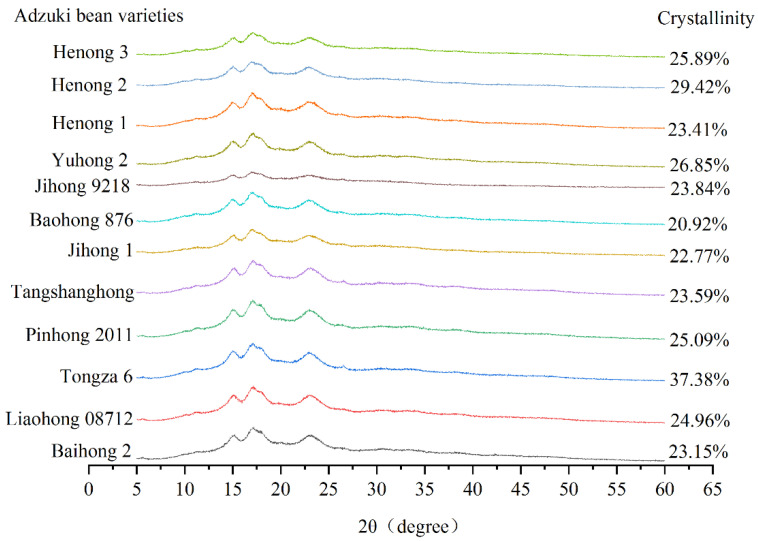
Starch crystallinity and X-ray diffraction patterns of different adzuki bean varieties.

**Figure 6 foods-11-03381-f006:**
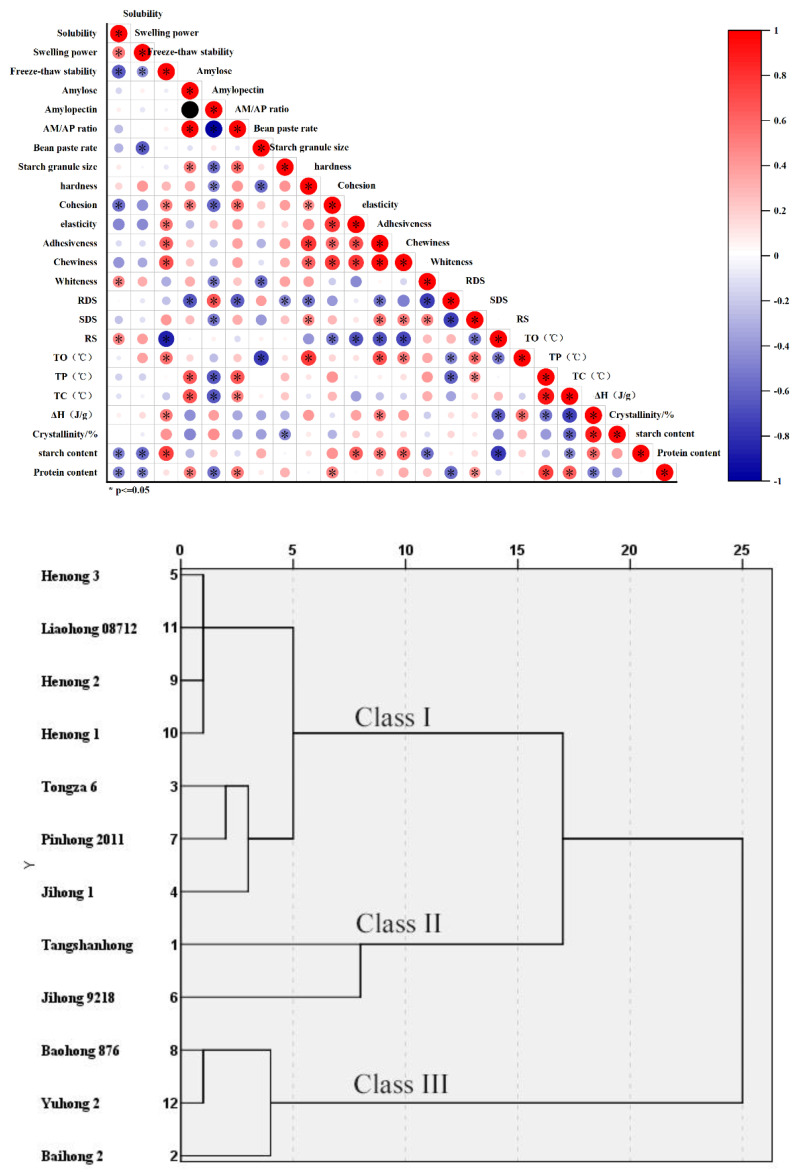
Correlation and cluster analysis of starch characteristic indices in different adzuki bean varieties.

**Table 1 foods-11-03381-t001:** Source and maturity of different adzuki bean varieties.

Variety Source	Variety Name	Breeding Unit	Latitude of Origin	Maturity
Different local varieties in China	Baihong 2	Baicheng Academy of Agricultural Sciences	45.6° N	precocious
	Liaohong 08712	Liaoning Academy of Agricultural Sciences	41.8° N	late maturing
	Tongza 6	Alpine Region Crop Research Institute, Shanxi Academy of Agricultural Sciences	40.1° N	late maturing
	Pinhong 2011	Chinese Academy of Agricultural Sciences	39.9° N	mid-late maturing
	Tangshanhong	Tangshan Academy of Agricultural Sciences	39.6° N	late maturing
	Jihong 1	Hebei Academy of Agriculture and Forestry Sciences	38.1° N	early maturing
	Baohong 876	Baoding Institute of Agricultural Sciences	38.8° N	precocious
	Jihong 9218	Hebei Academy of Agriculture and Forestry Sciences	38.1° N	mid-late maturing
	Yuhong 2	Chongqing Academy of Agricultural Sciences	29.5° N	precocious
	Henong 1	Hebei Agricultural University	43.2° N	precocious
	Henong 2	Hebei Agricultural University	42.9° N	precocious
	Henong 3	Hebei Agricultural University	42.7° N	precocious

**Table 2 foods-11-03381-t002:** Crude starch whiteness and texture characteristics of different adzuki bean varieties.

Variety		Measurement Indices				
	Whiteness	Hardness/N	Cohesion/Ratio	Elasticity/mm	Adhesiveness/N·s	Chewiness/mJ
Baihong 2	128.33 ± 0.551 ^d^	56.23 ± 8.54 ^d^	0.06 ± 0.01 ^f^	0.56 ± 0.13 ^f^	3.38 ± 1.86 ^f^	2.35 ± 0.13 ^e^
Liaohong 08712	131.27 ± 0.981 ^cd^	146.28 ± 6.62 ^bc^	0.14 ± 0.03 ^c^	1.57 ± 0.36 ^de^	18.27 ± 3.77 ^cde^	31.68 ± 2.34 ^cde^
Tongza 6	132.07 ± 1.914 ^cd^	154.02 ± 6.36 ^bc^	0.15 ± 0.03 ^bc^	1.88 ± 0.32 ^cd^	21.15 ± 1.85 ^cd^	51.21 ± 2.45 ^bcd^
Pinhong 2011	141.33 ± 0.289 ^a^	164.90 ± 10.45 ^bc^	0.15 ± 0.02 ^bc^	2.25 ± 0.36 ^bc^	24.90 ± 2.48 ^bc^	56.65 ± 2.61 ^bc^
Tangshanhong	135.03 ± 0.208 ^bc^	256.10 ± 12.03 ^a^	0.19 ± 0.01 ^a^	2.79 ± 0.31 ^a^	39.73 ± 4.46 ^a^	88.98 ± 3.45 ^a^
Jihong 1	118.03 ± 0.709 ^f^	173.78 ± 19.42 ^b^	0.15 ± 0.02 ^bc^	2.42 ± 0.19 ^ab^	27.73 ± 3.45 ^bc^	78.32 ± 5.45 ^ab^
Baohong 876	132.07 ± 1.779 ^bc^	133.52 ± 14.23 ^c^	0.12 ± 0.02 ^cd^	1.39 ± 0.46 ^e^	9.88 ± 2.22 ^ef^	15.25 ± 5.51 ^e^
Jihong 9218	140.17 ± 0.802 ^ab^	176.33 ± 12.66 ^b^	0.18 ± 0.02 ^ab^	2.56 ± 0.34 ^ab^	34.07 ± 3.19 ^ab^	80.79 ± 4.32 ^ab^
Yuhong 2	137.10 ± 0.300 ^bc^	72.05 ± 15.23 ^d^	0.08 ± 0.02 ^ef^	0.73 ± 0.19 ^f^	5.87 ± 1.07 ^f^	3.41 ± 1.35 ^e^
Henong 1	130.83 ± 0.513 ^cd^	77.38 ± 10.88 ^d^	0.09 ± 0.02 ^de^	1.21 ± 0.22 ^e^	8.97 ± 1.76 ^ef^	11.21 ± 2.12 ^e^
Henong 2	133.47 ± 0.709 ^cd^	137.90 ± 15.95 ^c^	0.14 ± 0.04 ^c^	1.51 ± 0.29 ^de^	13.42 ± 3.25 ^def^	19.84 ± 1.44 ^de^
Henong 3	121.93 ± 1.620 ^e^	150.33 ± 17.01 ^bc^	0.14 ± 0.01 ^c^	1.85 ± 0.57 ^cd^	19.27 ± 2.23 ^cde^	35.60 ± 3.32 ^cde^

Note: Different letters in the same column indicate differences under different treatments at *p* < 0.05 level, the same as below.

**Table 3 foods-11-03381-t003:** Determination of thermodynamic properties of starch from different adzuki bean varieties.

Variety	Onset Temperature T_O_ (°C)	Peak Temperature T_P_ (°C)	Conclusion Temperature T_C_ (°C)	Enthalpy ΔH (J/g)
Baihong 2	64.39 ± 0.9 ^abc^	71.66 ± 1.66 ^a^	84.39 ± 0.77 ^a^	2.78 ± 0.83 ^b^
Liaohong 08712	62.67 ± 1.54 ^bc^	69.76 ± 2.39 ^a^	83.50 ± 3.01 ^a^	2.79 ± 0.42 ^b^
Tongza 6	63.9 ± 1.31 ^abc^	72.12 ± 1.82 ^a^	82.43 ± 1.91 ^a^	5.80 ± 3.61 ^a^
Pinhong 2011	58.22 ± 5.88 ^bc^	69.92 ± 4.73 ^a^	82.27 ± 4.07 ^a^	3.17 ± 0.99 ^ab^
Tangshanhong	66.85 ± 3.85 ^ab^	78.39 ± 11.77 ^a^	92.71 ± 18.56 ^a^	2.32 ± 0.66 ^b^
Jihong 1	54.62 ± 9.68 ^c^	70.12 ± 1.32 ^a^	81.83 ± 0.20 ^a^	3.11 ± 0.77 ^ab^
Baohong 876	54.06 ± 9.05 ^c^	78.76 ± 15.53 ^a^	92.45 ± 22.18 ^a^	2.81 ± 0.80 ^b^
Jihong 9218	74.01 ± 13.59 ^a^	72.58 ± 16.33 ^a^	83.16 ± 18.16 ^a^	4.48 ± 3.05 ^ab^
Yuhong 2	57.02 ± 6.87 ^c^	75.65 ± 4.00 ^a^	87.80 ± 0.75 ^a^	2.54 ± 1.54 ^b^
Henong 1	62.35 ± 0.78 ^bc^	70.45 ± 0.95 ^a^	84.98 ± 3.35 ^a^	3.45 ± 0.67 ^ab^
Henong 2	60.84 ± 4.57 ^bc^	70.03 ± 0.82 ^a^	82.78 ± 1.31 ^a^	4.10 ± 1.19 ^ab^
Henong 3	64.03 ± 0.46 ^bc^	69.52 ± 0.01 ^a^	83.56 ± 0.41 ^a^	3.84 ± 0.54 ^ab^

Note: Different letters in the same column indicate differences under different treatments at *p* < 0.05 level.

## Data Availability

Not applicable.

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
