# Peer review of "Analysis and Research on Starch Content and Its Processing, Structure and Quality of 12 Adzuki Bean Varieties"

_foods, 2022, doi:10.3390/foods11213381_

Round 1

Reviewer 1 Report

The manuscript compares physiochemical properties of different varieties of red beans and groups them into classes via cluster analysis. The undertaken topic is interesting and the experiment design is proper. However there are several major and minor issues that should be addressed:

Line 40 – please use whole word vitamin

Line 41-42 high medicated health care function – not clear

Line 52-53 please provide some examples

Line 61-62 sentence not clear – highest RS content in total starch? Compared to what?

Line 70-75 it is not clears what the author intention was to present, moreover source should be provided

Line 85-87 please provide these source

Line 102 Please include the Table 1 of the supplementary material in the manuscript

Line 145 please add Chinese Standards and provided basic principle of the method as they are numerous methods for amylose determination and everyone has its limitations

Line 148 Please provide more detail on how whitness is calculated

Line 161 is would be more suitable to provide the speed in mm/s -> 1 mm/s

Line 163 was -> were

Line 174 symbols in equation 1 should be explained

Line 177 sovled?

Equation 4 – expanded starch?

Line 220 why so narrow temperature was choosen?

Line 232 what type of correlation was used and what method of grouping was used for cluster analsysis?

Line 259 it was not stated how granule size was determined

Line 272 please use term granule instead of grain

Line 287 figure 1b is in fact repetition of information given in figures 1a and partially 1c, figure 1d starch granule size should be in μm and it is not clear whether its mean, median and it was not discussed whether granule size distribution was unimodal

Line 292-296 the whiteness values will seem a little bit odd for most starch scientists as usually  CIE L value is used to characterize whiteness of starch, please discusse this information with regards to that

Line 309 Adhesiveness is usually given in N·s (please verify the unit)

Line 314-315 style

Line 344-345 style

Line 367 fig 2d – y axis has wrong title, moreover swelling power should be in g/g

Line 370 please either merge figure 3 in one or separate into 3

Line 372 style – crystalline

Line 391 – pleas include the difractograms in the supplementary material, while degree of crystanility should included in the manuscript.

Line 401 please move the figure 4 next to the discussion - subsection 3.7. Morovoer please enlarge the graphs – they are barely readable and color or highlight the clusters hat were indicated in the abstract

Line 429 – please use terms onset, peak and conclusion temperature

Line 432 ecological types?

Line 464 style

Line 491 please use term DSC conclusion temperature

Author Response

We are very grateful to the reviewers for their valuable comments.

We have revised the manuscript accordingly and revised sections are marked with revision mode. We hope this will make it more acceptable for publication.

Yours sincerely 

Yuechen Zhang Professor

Agricultural College, Hebei Agricultural University

2596 Lekai South Street, Baoding-071001, Hebei, P.R. China

Email: zhangyc1964@126.com

Tel: +86-312-7521499

Fax: +86-312-7521499

Reviewer 2 Report

The article is preapred correctly. Researches studied starch content, processing, digestion and structural Quality of 12 adzuki beans varieties. Researches focused on comparing the processing quality such as amylose and amylopectin content, whiteness, gel texture properties and other features of 12 Chinese  adzuki bean varieties.

                The introduction is correctly written and informative, fully revealing the research aspect.

                The materials and method is well described

                Resualts and analysis

The results are presented legibly, the discussion is carried out correctly, but...

In the description of the results, only the descriptive statistics were used. Why are there no comparisons? (line 242-257)

Please indicate which differences were statistically significant? (line 292-306)

Please check the results of statistical analyzes,  - Figure 3

Author Response

We are very grateful to the reviewers for their valuable comments.

我们对手稿进行了相应的修改,修改后的部分标有修改模式。我们希望这将使它更容易被出版所接受。

此致 

张月臣 教授

河北农业大学农学院

中国河北省保定市乐凯南街2596号-071001

邮箱:zhangyc1964@126.com

电话:+86-312-7521499

传真:+86-312-7521499

Reviewer 3 Report

Dear Editor and Authors,

This is an interesting paper. Authors have highlighted the potential application of starch content extracted from Adzuki Bean Varieties. I really appreciate the author’s effort in collection of samples and compiling the results of different starch sources extracted from Adzuki Bean Varieties. However, the language of the manuscript needs serous check. I have pointed out my specific points below.

Abstract

Write the importance/background of this study in the first sentence

Include the statistical results in the abstract

Keywords: Avoid the words used in the title

Introduction

L38-41: Add a reference for the statement L38-41

Add the reference for the statement L70-75. Refer the paper “The application of emerging non-thermal technologies for the modification of cereal starches” and “Ozone: An advanced oxidation technology for starch modification” for the same.

Write the novelty of this study

Materials and methods

Authors have done any study on estimation/evaluation of color profile of starch content

Results and discussion

Add more discussion part. Authors have highlighted the results. I suggested to add more scientific reason behind the results. Refer the recently published papers from MDPI-Foods.

Add the original image of the samples extracted from different source

Conclusion

Make it sharp. Please highlight only the significant findings

References

Update the references. Replace the old reference (published before 2017) with recent references

Author Response

(The authors gave the same response as above.)

Round 2

Reviewer 1 Report

Thank you for corrections and detailed answers. I have only one minor remark - please state what type od linkage method was used for CA, i.e. single linkage, Ward's method etc.

Author Response

We are very grateful to the reviewers for their valuable comments.
We have revised the manuscript accordingly and revised sections are marked with revision mode. We hope this will make it more acceptable for publication. 
Yours sincerely  

Reviewer 2 Report

The manuscript has been sufficiently improved. Thank You.

Author Response

(The authors gave the same response as above.)

Reviewer 3 Report

The revised version is suitable for publication 

Author Response

(The authors gave the same response as above.)
